# Bacterial polyphosphates interfere with the innate host defense to infection

Julian Roewe [1], Georgios Stavrides [1], Marcel Strueve[1], Arjun Sharma [1,2], Federico Marini [1,3], Amrit Mann[1], Stephanie A. Smith[4], Ziya Kaya [5], Birgit Strobl [6], Mathias Mueller [6], Christoph Reinhardt [1], James H. Morrissey [4] & Markus Bosmann [1,2✉]

Polyphosphates are linear polymers and ubiquitous metabolites. Bacterial polyphosphates are long chains of hundreds of phosphate units. Here, we report that mouse survival of peritoneal *Escherichia coli* sepsis is compromised by long-chain polyphosphates, and improves with bacterial polyphosphatekinase deficiency or neutralization using recombinant exopolyphosphatase. Polyphosphate activities are chain-length dependent, impair pathogen clearance, antagonize phagocyte recruitment, diminish phagocytosis and decrease production of iNOS and cytokines. Macrophages bind and internalize polyphosphates, in which their effects are independent of P2Y1 and RAGE receptors. The M1 polarization driven by *E. coli* derived LPS is misdirected by polyphosphates in favor of an M2 resembling phenotype. Long-chain polyphosphates modulate the expression of more than 1800 LPS/TLR4-regulated genes in macrophages. This interference includes suppression of hundreds of type I interferon-regulated genes due to lower interferon production and responsiveness, blunted STAT1 phosphorylation and reduced MHCII expression. In conclusion, prokaryotic polyphosphates disturb multiple macrophage functions for evading host immunity.

[1] Center for Thrombosis and Hemostasis, University Medical Center Mainz, 55131 Mainz, Germany. [2] Pulmonary Center, Department of Medicine, Boston University School of Medicine, Boston, MA 02118, USA. [3] Institute of Medical Biostatistics, Epidemiology and Informatics, University Medical Center Mainz, 55131 Mainz, Germany. [4] Department of Biological Chemistry, University of Michigan Medical School, Ann Arbor, MI 48109-1085, USA. [5] Department of Medicine III, University of Heidelberg, 69120 Heidelberg, Germany. [6] Institute of Animal Breeding and Genetics, Department of Biomedical Science, University of Veterinary Medicine Vienna, 1210 Vienna, Austria. ✉email: mbosmann@bu.edu

Bacterial infections cause a major burden of disease world-wide[1]. It is estimated that more than 50 million people contract severe infections each year[2,3]. The therapeutic options for the life-threatening complications of these infections are limited, demanding better insights into the intricate relationship between pathogens and the host[4].

*Escherichia coli* (*E. coli*) is a pathogenic bacterium infecting sterile anatomic sites of mammalian hosts, where it encounters surveilling TLR4-expressing macrophages as a first line of immune defense[5,6]. *E. coli* can produce high-molecular weight, long-chain polyphosphates (>300−1000 $P_i$) from ATP by poly-phosphatekinases (Ppk) in a reversible enzymatic reaction balanced by polyphosphatases (Ppx) (Fig. 1a)[7,8]. *Escherichia coli* store polyphosphates in specialized conserved organelles termed metachromatic polyphosphate granules or acidocalcisomes[9–11]. In some bacteria, polyphosphates are also localized on the cell surface as capsule-like coatings[11]. Microbial polyphosphate synthesis accelerates during periods of stress and starvation, when polyphosphates may accumulate to 10–20% of dry weight of the cells[11–13]. The biological functions of inorganic polyphosphates in living organisms are pleiotropic and mysterious. Polypho-sphate metabolites serve as an energy reservoir and phosphate reserve. In *E. coli*, polyphosphates participate in bacterial fitness, stress resistance, regulation of prokaryotic transcription, transla-tion, act as protein-folding scaffolds and primordial chaperones[8,14].

In mammalian organisms, major pools of endogenous poly-phosphates are present as low molecular weight, short chains

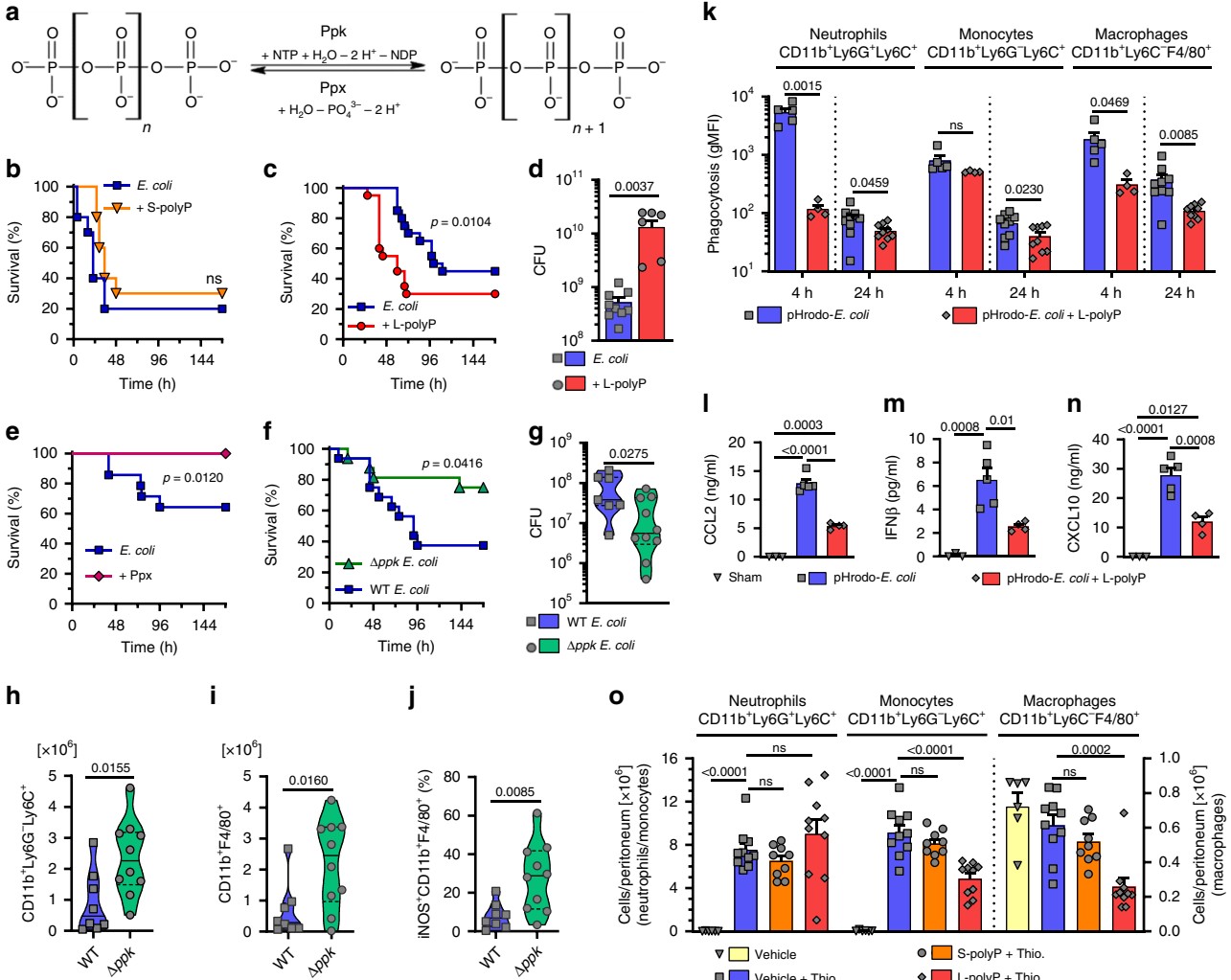

**Fig. 1 Bacterial polyphosphates impair sepsis survival and myeloid cell responses. a** Bacterial polyphosphate metabolism, Ppk polyphosphatekinase, Ppx exopolyphosphatase. **b, c** Mice were injected i.p. with *E. coli* ± short-chain polyphosphates (S-polyP) (**b** 10 µg/g, $n = 10$ mice/group) or *E. coli* ± long-chain polyphosphates (L-polyP) (**c** 10 µg/g, $n = 20$ mice/group). **d** Mice were infected with *E. coli* alone ($n = 9$) or *E. coli* + L-polyP ($n = 7$) for 24 h and peritoneal colony-forming units (CFU) were determined. **e** Survival after infection with *E. coli* ($n = 14$ mice) or *E. coli* + recombinant Ppx (2 µg/g, $n = 15$ mice). **f** Germ-free mice were gavaged for intestinal monocolonization with wild-type (WT) *E. coli* or Ppk-deficient (Δ*ppk*) *E. coli* followed after 2 weeks by CLP sepsis ($n = 16$ mice/group). **g−j** Monocolonization (as in **f**) with WT *E. coli* ($n = 8$ mice) or Δ*ppk* *E. coli* ($n = 10$ mice) followed by CLP sepsis and isolation of peritoneal lavages after 48 h for analysis of total counts of CFU (**g**), monocytes (**h**), macrophages (**i**), or frequencies of iNOS[+] macrophages (**j**). **k** Phagocytosis of injected i.p. pHrodo-*E. coli* by peritoneal neutrophils, monocytes, and macrophages ± L-polyP after 4 h ($n = 5$ *E. coli*, $n = 4$ *E. coli* + L-polyP) or 24 h ($n = 9$ mice/group). **l−n** CCL2 (**l**), IFNβ (**m**), and CXCL10 (**n**) in plasma 4 h after pHrodo-*E. coli* ± L-polyP i.p., ELISA ($n = 3$ Sham, $n = 5$ *E. coli*, $n = 4$ *E. coli* + L-polyP). **o** Thioglycolate (Thio.) ± S-/L-polyP induced migration of myeloid cells to peritoneal compartment of mice after 24 h, vehicle: phosphate buffered saline (PBS) ($n = 6$ vehicle, $n = 10$ Thio., $n = 10$ Thio. + L-polyP, $n = 9$ Thio. + S-polyP). **h−k, o** flow cytometry. Data are expressed as mean ± s.e.m., **d, g−i** median is indicated as solid line and quartiles (1st, 3rd) as dotted lines, **b, c** two-sided Gehan−Breslow−Wilcoxon test, **e, f** two-sided Log-rank (Mantel −Cox) test, **d, g−k** two-sided *t* test, **l−o** one-way ANOVA, ns not significant ($p > 0.05$).

(~50−150 $P_i$), stored in secretory dense granules of platelets and mast cells[7]. Polyphosphates are released in responses to injury and cellular activation for the modulation of proteolytic plasma cascades of coagulation, bradykinin and complement[7,15−19]. The effects and potency of polyphosphates are dependent on their polymer size[17]. Recent reports have suggested that short-sized, mammalian-derived polyphosphates regulate the functions of endothelial cells and astrocytes by signaling through the purinergic P2Y1 receptor and RAGE receptor[20,21]. Short-chain polyphosphates can modulate the migration, differentiation and functions of neutrophils and macrophages[22−24].

The rationale for the presented work is to better define the role of bacterial polyphosphates as a variable of host−pathogen interactions. We uncover harmful activities of long-chain polyphosphates in live *E. coli* infection. Neutralization or bacterial deficiency of polyphosphates results in better survival rates of infected mice. Polyphosphates act in strict dependency of chain length and antagonize the local recruitment of professional phagocytes. In cultured macrophages, long-chain polyphosphates interfere with M1 polarization, suppress iNOS, antagonize type I interferon responses together with decreased STAT1 activation, and impede MHC II expression. We conclude that bacterial polyphosphates represent a pathogen evasion strategy by disturbing the natural macrophage response. The neutralization of polyphosphates has therapeutic potential to mitigate immunosuppression and to restore innate host defense to bacterial infection.

## Results

**Bacterial polyphosphates mediate death in *E. coli* sepsis**. To investigate the relevance of polyphosphates in infection, live *E. coli* were injected together with chemically synthesized polyphosphates of either short-chain sizes ("mammalian"; S-polyP, $n \sim P_i 70$) or long-chain sizes ("bacterial"; L-polyP; $n \sim P_i 700$) (Fig. 1b, c, Supplementary Fig. 1a). While no significant change of survival was observed with S-polyphosphates (Fig. 1b), the L-polyphosphates resulted in accelerated mortality (Fig. 1c) and a higher bacterial burden in the peritoneal cavity (Fig. 1d). Next, we purified a highly active, recombinant Ppx (from *Saccharomyces cerevisiae*[14]) for interventional degradation of bacterial-derived polyphosphates (Supplementary Fig. 1b). The treatment with Ppx improved the survival of live *E. coli* sepsis by a margin of 36% between the control group (64% survival) and the Ppx group (100% survival; Fig. 1e). Direct microbicidal effects of exogenous L-polyphosphates and Ppx on *E. coli* growth in LB medium were excluded (Supplementary Fig. 1c).

The magnitude of endogenous L-polyphosphate accumulation in *E. coli* is a function of environmental conditions[8,14]. To model a natural habitat for *E. coli* followed by host infection, germ-free mice were monocolonized with either *E. coli* wild type (WT) or a previously characterized Ppk-deficient (*Δppk*) *E. coli* strain[14] (Supplementary Fig. 1c−f). After 2 weeks, the monocolonized groups of mice were subjected to the cecal ligation and puncture (CLP) procedure for induction of monomicrobial sepsis. The peritoneal infection with *E. coli Δppk* following CLP resulted in better survival rates (38% vs. 75%; Fig. 1f), reduced the colony-forming units (CFUs) (Fig. 1g) and lower local polyphosphate concentrations (Supplementary Fig. 1g) as compared to gnotobiotic mice with *E. coli* WT monocolonization. The peritoneal influx of CD11b+Ly6G−Ly6C+ monocytes and CD11b+F4/80+ macrophages (Fig. 1h, i; Supplementary Figs. 1h, i, 2a) including their iNOS expression (Fig. 1j) was inversely correlated with polyphosphate amounts (Supplementary Fig. 1g). In addition, CD11b+Ly6G+Ly6C+ neutrophil viability was better preserved during infection with *E. coli Δppk* (Supplementary Fig. 1j). In

synopsis, these findings suggested that *E. coli*-derived L-polyphosphates interfered with an efficient anti-microbial myeloid cell response. In fact, when fixed *E. coli* conjugated with a pH-sensitive fluorescence reporter were injected intraperitoneally into conventionally housed mice, the phagocytosis of such particles by neutrophils, monocytes and macrophages was reduced by addition of chemically synthesized L-polyphosphates (Fig. 1k, Supplementary Figs. 2b, 3a−c). Moreover, L-polyphosphates suppressed the release of macrophage attracting chemokines (CCL2, CXCL10) and the CXCL10 inducing cytokine, IFNβ (Fig. 1l−n). L-polyphosphates also reduced the recruitment and maturation of CD11b+Ly6C+ monocytes (increase of Ly6C marker; Supplementary Fig. 3d, e) and decreased CD11b+F4/80+ macrophage numbers in response to thioglycolate i.p., while no consistent effects were seen for the short-chain S-polyphosphates (Fig. 1o, Supplementary Fig. 2c). Altogether, these findings suggested that L-polyphosphates impede the innate host defense and increase lethality of bacterial sepsis.

**Polyphosphates modulate the phenotypes of macrophages**. To further study the direct effects of polyphosphates on macrophage immune responses in exclusion of potential confounders (e.g. alterations in *E. coli* phenotype[14,25,26] or plasma protease activation[15,18,27,28]), we turned our focus to serum-free cultured C57BL/6J macrophages and treated them with *E. coli*-derived LPS and synthetic polyphosphates. The amounts of polyphosphates are expressed as monophosphate unit concentrations. The S-/L-polyphosphates did not negatively affect macrophage viability (Supplementary Fig. 4). L-polyphosphates avidly bound to the surface of CD11b+F4/80+ macrophages within 30 min (Fig. 2a, b, Supplementary Fig. 2d), when incubated on ice to prevent internalization. This binding was antagonized by a 15-min preincubation of macrophages with dextran sulfate. Dextran sulfate is a polyanionic substance for saturating and masking the positively charged surface molecules including scavenger receptors on macrophages. Thus, the blockade of polyphosphate binding by dextran sulfate suggested that the initial interaction of polyphosphates with macrophages relied on electrostatic forces. Next, we probed macrophages at 37 °C with biotin-labeled L-polyphosphates for live cell imaging. Macrophages internalized the polyphosphates in colocalization with the endosomal membrane compartment under the applied conditions (Fig. 2c).

To screen for the transcriptome-wide effects of polyphosphates on host defense pathways, we performed RNA-seq. L-polyphosphates alone resulted in 749 significantly downregulated differentially expressed genes (DEGs) and 736 upregulated DEGs at the 12 h time point (Supplementary Fig. 5a). In combination with LPS, the L-polyphosphates suppressed 1083 DEGs and increased 814 DEGs as compared to LPS alone (Supplementary Fig. 5b). The bioinformatics analyses identified several over-represented cellular pathways affected by L-polyphosphates including cellular responses to IL-4 and IFNγ, macrophage differentiation and regulation of leukocyte activation (Supplementary Figs. 5c, d, 6).

The modulation of the aforementioned pathways by L-polyphosphates suggested that the polarization of macrophages might be affected. While an imperfect schematization, macrophage phenotypes are often classified into M1 or M2 states[6,29]. LPS promotes an M1 macrophage polarization as part of the normal defense response to infection[6]. When DEGs were grouped as either M1 or M2[29,30], L-polyphosphates tended to enhance the expression of M2 genes in LPS-activated macrophages, while antagonizing M1 genes (Fig. 3a). An important gene associated with M1 phenotype is *iNOS* (*Nos2*), whose mRNA abundance

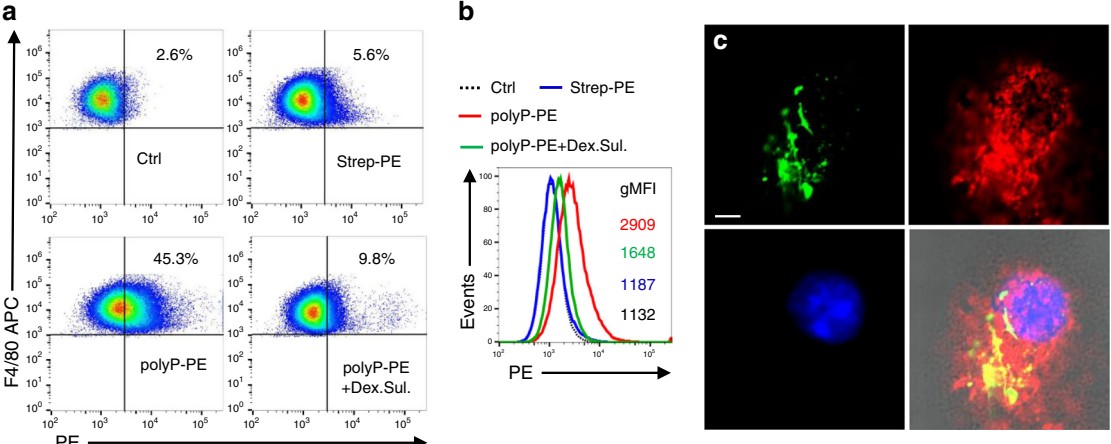

**Fig. 2 Macrophages bind and internalize polyphosphates. a** CD11b[+]F4/80[+] macrophages (BMDM) from C57BL/6J mice were incubated with biotinylated L-polyphosphates (50 μM) for 30 min on ice followed by streptavidin-PE (PolyP-PE). The negative controls were unstained (Ctrl) or streptavidin-PE (Strep-PE) only. The preincubation with dextran sulfate (Dex. Sul., 100 μg/ml) for 15 min reduced the binding of polyphosphates to macrophages; flow cytometry with pre-gating on CD11b[+] cells (CD11b-Pacific Blue). **b** Representative overlay histograms from the same experiments as shown in frame (**a**) and numbers of geometric mean fluorescence intensities (gMFI). **c** Live cell imaging of macrophages incubated with biotinylated L-polyphosphates plus streptavidin-FITC (green), CellMask Orange membrane-staining dye (red) and nuclear stain with Hoechst (blue), Objective ×63, scale bar: 5 μm. Data are representative of three (**a**, **b**) and two (**c**) independent experiments.

was also substantially suppressed by L-polyphosphates but less consistently affected by S-polyphosphates (Fig. 3b). L-polyphosphates counteracted the LPS-induced presence of intracellular iNOS protein (Fig. 3c, d, Supplementary Fig. 5e) and $NO_2^-$ release in supernatants as a surrogate of nitric oxide formation (Fig. 3e).

RAGE and P2Y1 have been suggested as putative receptors for polyphosphates in non-leukocytic cells[20,21]. To test the hypothesis if polyphosphates antagonize iNOS expression of LPS/TLR4-activated macrophages through these receptors, we compared $Rage^{-/-}$ and $P2y1^{-/-}$ derived macrophages to wild-type macrophages (Fig. 3f, g). In fact, polyphosphates did not require RAGE and P2Y1 receptors to mediate their effects in macrophages (Fig. 3f, g). On the other hand, the iNOS induction by LPS required IFNβ (Supplementary Fig. 5f). The prominent M2 marker, CD206, was induced by L-polyphosphates alone and further augmented when L-polyphosphates were combined with recombinant IL-4 (Fig. 3h, i, Supplementary Fig. 5g). In addition, IL-4-induced arginase-1 (Arg1) was increased by L-polyphosphate co-stimulation (Fig. 3j). However, some M2 markers such as Ym1 and Fizz1 were suppressed by L-polyphosphates (Fig. 3a, j), suggesting that the effects of bacterial L-polyphosphates are not strictly M2, but may suffice for preventing an efficient, anti-microbial M1 macrophage phenotype.

**Polyphosphates reprogram the type I interferon response.** The RNA-seq pathway analysis showed strong effects of L-polyphosphates regarding interferon responses in LPS/TLR4-activated macrophages (Supplementary Figs. 5d, 7). In fact, interferon-regulated genes[31] (IRGs; ↓n = 738, ↑n = 481) represented the majority (64.3%) of all L-polyphosphate-dependent DEGs (n = 1897) and the most pronounced quantitative differences were seen for type I IRGs (Fig. 4a, b, Supplementary Fig. 5d). As a general trend, most IRGs were down-modulated by L-polyphosphates (Fig. 4a). This observation was emphasized by impaired LPS-induced transcription of prominent IRGs such as Ifi44, Irf7 and Ifit1 observed during subsequent RT-qPCR studies (Fig. 4c). L-polyphosphates dose-dependently suppressed CXCL10 chemokine release, while S-polyphosphates mediated a

trend for a minor reciprocal effect (Fig. 4d). As anticipated, the biological activities of S-/L-polyphosphates were completely lost following their digestion by phosphatase (Fig. 4e). To study the specificity of L-polyphosphate interference with different bacterial TLR pathways, we compared agonists for TLR2 (Zymosan and Peptigolycan), TLR4 (LPS) and TLR9 (dsDNA) (Fig. 4f).

The reduction of CXCL10 and other IRGs was explained by lower IFNβ release in LPS/TLR4-activated macrophages (Fig. 4g). Furthermore, L-polyphosphates (but not S-polyphosphates) antagonized LPS-induced phosphorylation of STAT1[Y701] in CD11b[+]F4/80[+] macrophages as studied by flow cytometry (Fig. 4h). In western blots, L-polyphosphates reduced the ratio of phospho-STAT[Y701]/STAT1 (Fig. 4i, j, Supplementary Fig. 8) as a critical event regulating the induction of type I interferons. L-polyphosphates alone did not induce phospho-STAT1[Y701] or alter total STAT1 protein after 3 h (Supplementary Fig. 8). In $Stat1^{-/-}$ macrophages, LPS lost much of its activity for induction of iNOS gene expression and residual iNOS (Nos2) mRNA was not significantly inhibited by L-polyphosphates (Supplementary Fig. 9a). Disruption of type I interferon signaling using $Ifnar1^{-/-}$ macrophages mainly nullified LPS and L-polyphosphates-dependent effects for iNOS (Nos2), Stat1 and Cxcl10 (Supplementary Fig. 9b–d). In addition to decreased IFNβ release (Fig. 4g), L-polyphosphates also diminished the responsiveness of wild-type macrophages to IFNβ (Fig. 4k). In detail, when macrophages were stimulated with recombinant IFNβ, L-polyphosphates (but not S-polyphosphates) consistently suppressed IRGs such as Cxcl10, iNOS (Nos2) and Msc4a4c (Fig. 4k, l, Supplementary Fig. 5f). The enhanced induction of the Jak-STAT pathway inhibitor, Socs1 (Fig. 4c), in the presence of L-polyphosphates might explain this decreased interferon responsiveness. In summary, polyphosphates were unique regulators of the STAT1-dependent host interferon response.

**Interference of polyphosphates with antigen presentation.** An important function of macrophages is professional antigen presentation. We identified the major histocompatibility complex pathways (MHCI, MHCII) as overrepresented in the RNA-seq data sets of LPS ± L-polyphosphates treated macrophages (Fig. 5a, Supplementary Fig. 5d). RT-qPCR experiments revealed that

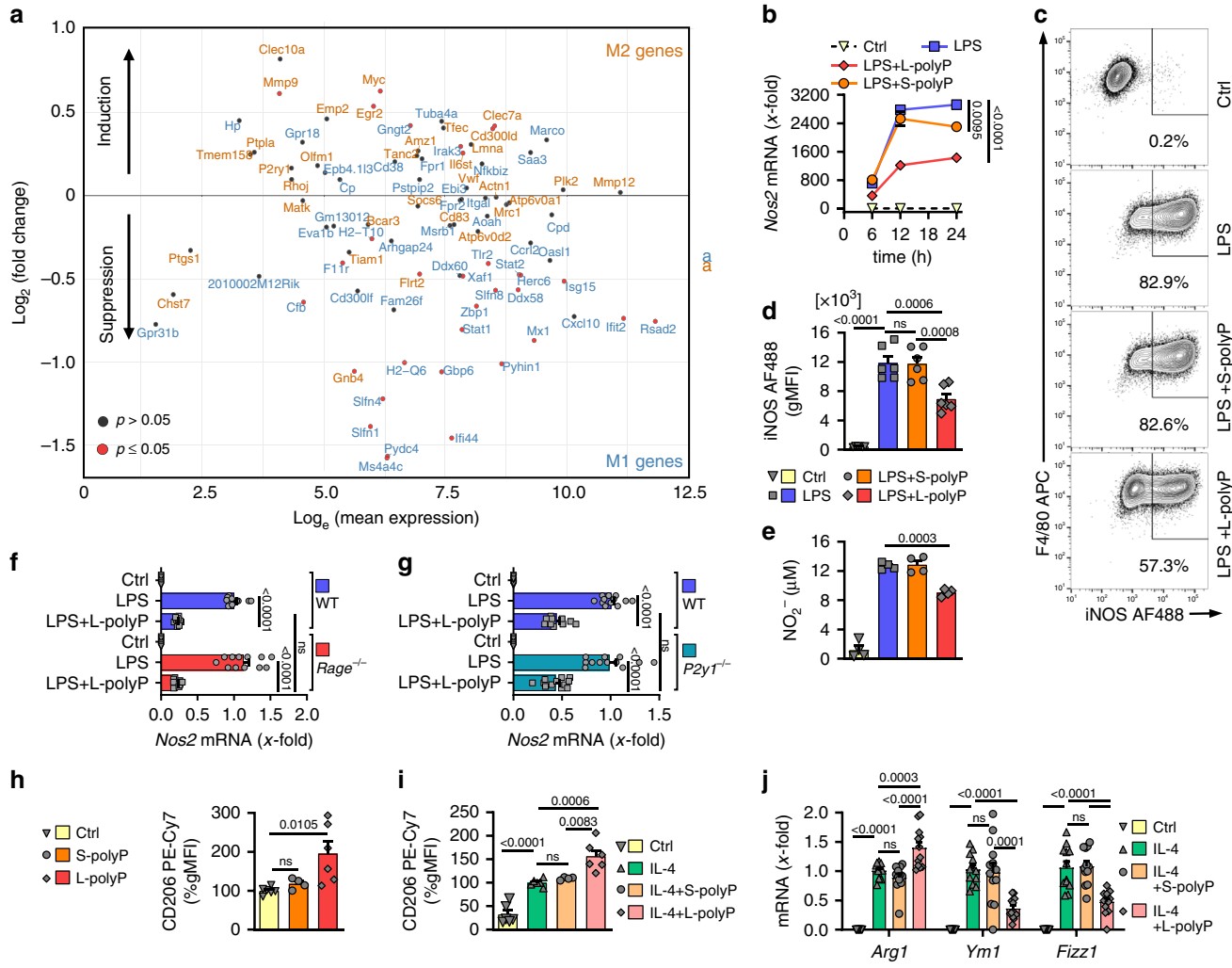

**Fig. 3 Polyphosphates attenuate LPS-induced M1 macrophage polarization in favor of increased M2 characteristics. a** RNA-seq analysis of M1 (blue) and M2 (orange) markers in macrophages (BMDM, C57BL6/J) stimulated for 12 h with LPS ± long-chain polyphosphates (L-polyP, $n = 5$/group). Differentially expressed genes (adjusted $p$ value ≤ 0.05) of LPS + L-polyP vs. LPS alone are indicated as red dots. **b** Time course of iNOS (Nos2) mRNA as archetype M1 marker induced by LPS ± S-/L-polyP and untreated controls (Ctrl), RT-qPCR ($n = 4$ samples/group for each time point). **c** Intracellular iNOS induced by LPS ± S-/L-polyP after 24 h in macrophages by representative flow cytometry pre-gated on CD11b⁺ cells (CD11b-Pacific Blue). **d** Intracellular iNOS as geometric mean fluorescence intensities (gMFI) in CD11b⁺F4/80⁺ macrophages from groups of mice ($n = 6$) as in frame (**c**). **e** $NO_2^-$ concentrations in macrophage supernatants as surrogate for NO production after LPS ± S-/L-polyP, 24 h ($n = 4$ samples/group). **f** Rage⁻/⁻ and WT macrophages were compared for iNOS (Nos2) mRNA after LPS ± L-polyP. The expression values of LPS-stimulated WT macrophages were set as onefold for normalization, 12 h, RT-qPCR ($n = 12$/group). **g** P2y1⁻/⁻ macrophages and WT macrophages analyzed for iNOS (Nos2) expression after LPS ± L-polyP, 12 h, RT-qPCR ($n = 12$/group). **h, i** Expression of the M2 marker, CD206, as determined by intracellular flow cytometry and expressed as normalized gMFI. Macrophages were incubated with S-polyP or L-polyP for 24 h (**h**) or preincubated 24 h with IL-4 before 24 h stimulation with S-/L-polyP (**i**), **h, i** $n = 6$/group exept where S-polyP ($n = 4$). **j** Expression of M2 markers Arg1, Ym1, and Fizz1 by BMDM stimulated 48 h with IL-4 ± S-/L-polyP, RT-qPCR ($n = 12$/group). Data are expressed as mean ± s.e.m. and representative (**b, e**) or combined (**d, f–j**) as examined over three independent experiments, **b** two-way ANOVA ($F(1,6)_{LPS\ vs.\ LPS+L-polyP} = 754.3$ and $F(1,6)_{LPS\ vs.\ LPS+S-polyP} = 14.06$), **d–j** one-way ANOVA, ns not significant ($p > 0.05$).

L-polyphosphates suppressed the MHC-inducing transcription factors Nlrc5, Ciita and Rfx5 (Fig. 5b). Accordingly, L-polyphosphates reduced the LPS-induced cell surface expression of MHCII invariant chain (CD74, I-A/I-E) on CD11b⁺F4/80⁺ macrophages (Fig. 5c, Supplementary Fig. 10a). The expression of costimulatory proteins CD80 and CD86 was also moderately toned down (Fig. 5d, e). S-polyphosphates did not down-modulate the surface expression of MHCII as was observed with L-polyphosphates (Supplementary Fig. 10b). Finally, a reduction of the MHCII invariant chain on macrophages was also observed in vivo after intraperitoneal injection of fluorescent E. coli particles ± L-polyphosphates into C57BL/6J mice (Fig. 5f, Supplementary Fig. 10c). Hence, bacterial polyphosphates appear to also defuse the interfaces linking innate with adaptive immunity.

## Discussion

In conclusion, our study reveals that long-chain polyphosphates are harmful bacterial metabolites during E. coli infection and interfere with the ensuing immune responses of macrophages on multiple different levels (Fig. 6). The synergistic consequences most likely provide an edge for the microbe to escape from host defenses.

We reveal that long-chain polyphosphates suppress the LPS/TLR4-induced release of IFNβ and reduce interferon responsiveness

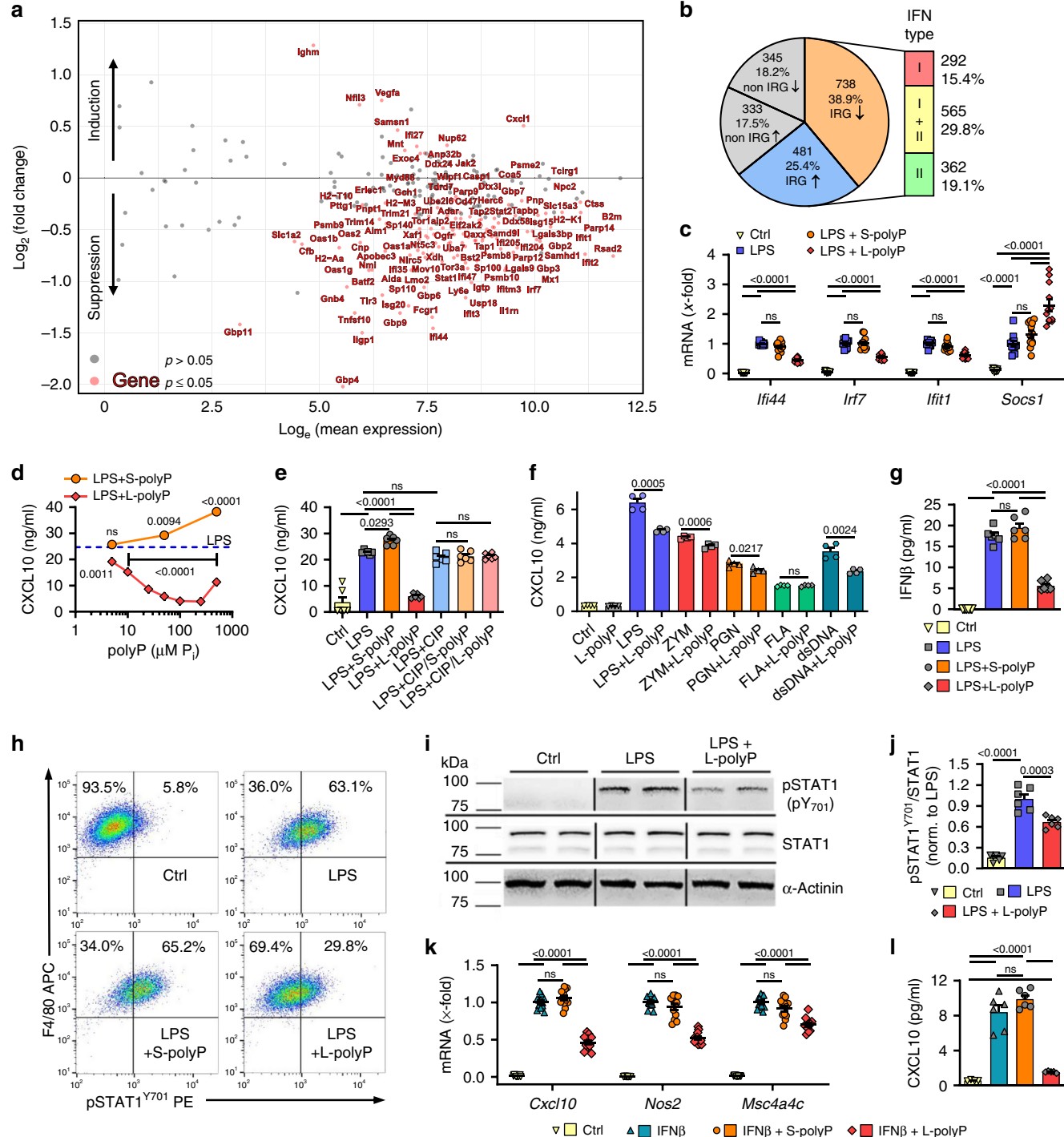

through inhibition of STAT1 phosphorylation. This interception ultimately affects many interferon-regulated genes, including CXCL10 and MHCII antigen presentation. Whereas it remains controversial (or at least context dependent) if type I interferons are beneficial for healing of bacterial infections[32,33], this notion may be reconciled by our findings that long-chain polyphosphates only act as specific modulators for subsets of IRGs.

Macrophages are essential for protection against *E. coli* infection[34]. We uncover that long-chain polyphosphates target macrophage differentiation and their functions such as iNOS expression. Myeloid-cell-derived iNOS is important for both *E. coli* clearance and M1 macrophage polarization[35,36]. Several bacterial pathogens have evolved strategies to interfere

with macrophage polarization[37]. The differentiation of human macrophages from peripheral blood also appears regulated by platelet-derived, short-chain polyphosphates[24]. In addition to our oberservation that long-chain polyphosphates decrease local macrophage recruitment, it was recently reported that short-chain ($P_i \sim 150$) polyphosphates block the migration of macrophages and their intracellular actin polarization together with effects on cyclooxygenase-2, TNFα and the JNK/p38 pathway[23]. In contrast to our findings of long-chain polyphosphates mediating detrimental effects in live *E. coli* sepsis, the short-chain ($P_i \sim 150$) polyphosphates protected from LPS-induced shock lethality[23]. This discrepancy supports the view that polyphosphate effects are variable for

**Fig. 4 Polyphosphate antagonism of type I interferons. a** Transcriptome modulation of interferon-regulated genes (IRGs) by long-chain polyphosphates (L-polyP) in LPS/TLR4-activated macrophages (C57BL/6J, BMDM), 12 h, $n = 5$ samples/group, RNA-seq. Differentially expressed genes (adjusted $p$ value ≤ 0.05) of LPS + L-polyP vs. LPS are indicated as red dots and tagged with their gene name. LPS alone is represented by baseline 0 on the y-axis/$\log_2$. **b** Overrepresentation of IRGs within all identified DEGs in RNA-seq data from LPS + L-polyP vs. LPS alone. **c** RT-qPCR of typical ISGs (Ifi44, Irf7 and Ifit1, 12 h) and inhibitory Socs1 (24 h) ($n = 12$/group except $n = 10$ for Ifit1 LPS + S-polyP). **d** CXCL10 in supernatants of macrophages incubated for 24 h with LPS ± increasing concentrations of S-/L-polyP ($n = 6$/group and concentration), ELISA. The blue dotted line represents CXCL10 release with LPS alone. **e** Release of CXCL10 from LPS/TLR4-activated macrophages ± calf intestinal phosphatase (CIP) digestion of S-/L-polyP ($n = 6$/group), 24 h, ELISA. **f** CXCL10 released by macrophages incubated with agonists for TLR4 (LPS), TLR2 (Zymosan [ZYM] and Peptidoglycan [PGN]), TLR5 (Flagellin [FLA]) and TLR9 (double-stranded DNA [dsDNA]) alone or combined with L-polyP, 24 h ($n = 4$/group). **g** IFNβ release by macrophages incubated with LPS ± S-/L-polyP ($n = 6$/group), 4 h, ELISA. **h** STAT1$^{Y701}$ phosphorylation in CD11b$^{+}$F4/80$^{+}$ macrophages 3 h after LPS ± S-/L-polyP, flow cytometry pre-gated on CD11b$^{+}$ cells (CD11b-Pacific Blue). **i** Phosphorylated STAT1$^{Y701}$ (top), total STAT1 (center), and α-Actinin (bottom) of macrophages incubated with LPS ± L-polyP for 3 h or kept untreated (Ctrl), western blotting of technical duplicates. **j** Densitometry and ratios of phosphorylated STAT$^{Y701}$/STAT1 protein normalized to signals with LPS alone ($n = 6$/group) as in frame (**i**). **k** Cxcl10, iNOS (Nos2), and Msc4a4c from macrophages preincubated for 3 h ± S-/L-polyP followed by 6 h rmIFNβ stimulation ($n = 12$/group), RT-qPCR. **l** CXCL10 in supernatants of macrophages kept as controls (Ctrl, $n = 4$) or preincubated for 3 h with rmIFNβ ($n = 6$) ± S-polyP ($n = 6$) or L-polyp ($n = 5$), 24 h, ELISA. Data are expressed as mean ± s.e.m. and representative (**h, i**) or combined (**c–e, g, j–l**) as examined over three independent experiments, or combined (**f**) from two of three independent experiments. **c, e, j–l** One-way ANOVA, **d, f** two-sided $t$ test vs. TLR agonist alone, ns not significant ($p > 0.05$).

chain length, dose and the model of infection. LPS-induced shock in mice is a rather poor model for human sepsis[38].

We found that long-chain polyphosphates avidly bind and are internalized by macrophages. The anionic polyphosphates can interact with hundreds of eukaryotic proteins[39,40], which impedes the allocation of a solitary molecular mechanism of action. Polyphosphates form complexes with the platelet-derived chemokine CXCL4[41]. While P2Y1 and RAGE were proposed as putative polyphosphate receptors by RNA interference studies in human and rat non-immune cells[20,21], P2Y1-deficient and RAGE-deficient macrophages were equally susceptible for polyphosphate-mediated suppression of iNOS (Nos2) (Fig. 3f, g). Thus, our data argue against RAGE and P2Y1 representing universal receptors for polyphosphates. It remains speculative, if the RNAi gene silencing was confounded by off-target effects[20,21,42]; or if polyphosphates react in a cell type-specific and species-specific manner. The higher biologic potency of long-chain polyphosphates (as opposed to short-chains) described in our studies could be reconciled with the idea, that long chains are required to assemble several protein subunits together as a functional multimeric complex in macrophages.

Another open question remains how bacteria-derived polyphosphates are liberated from E. coli to allow direct contact with macrophage proteins. E. coli contain intracellular polyphosphates, but it is unclear if they also expose polyphosphates loosely associated to their cell surface as was reported for Neisseria gonorrhoeae[43].

While an active release of short-chain polyphosphates exists for platelets and mast cells, no active secretion mechanism has been described so far for bacteria. The dissociation of prokaryotic polyphosphates could be a passive process as a consequence of damage and cell death of bacteria. In the course of infection, bacteria are destroyed by the versatile weaponry of the host immune defenses including nitric oxide, reactive oxygen species, toxic granules, lysozyme and alkalic phosphatase[44]. All these effector mechanisms could contribute to the release of polyphosphates from either extracellular bacteria or after their digestion in phagolysosomes. A current barrier is the absence of highly sensitive and chain-length-specific polyphosphate detection methods. We have used the only commercially available chromogenic assay to confirm abundance of polyphosphates in cell-free peritoneal lavage fluids during E. coli WT/Δppk infection (Supplementary Fig. 1g), but this assay is nonselective for chain length. The WT E. coli Ppk enzyme has been demonstrated to produce long-chain polyphosphates with molecular sizes of more

than >300 $P_i$ residues[45]. While ppk gene mutations could affect the amount of polyphosphate accumulation in E. coli, the variations in polyphosphate chain length were found to be only modest[45]. This suggests that long-chain polyphosphates are predominant during E. coli infection.

To underscore the functional distinctions inherent in polyphosphate polymer sizes, mammalian-type, short-chain polyphosphates were less active or showed divergent properties for some readouts of inflammation[23]. Intriguingly, high concentrations and direct secretion of relatively long-chain polyphosphates in mammalian species is merely reported for the immune privileged brain through astrocytes[20,46]. It would be interesting to study whether mammalian long-chain polyphosphates exert suppressive effects on the microglia in the absence of pathogens in this highly regulated tissue-specific immune milieu.

L-polyphosphate synthesis as high molecular weight polymers is a ubiquitous trait of bacteria and their effects on immunity might be highly conserved and relevant to diverse types of mammal-bacteria interactions. On the other hand, the enzymatic machinery for polyphosphate synthesis is less conserved across different bacteria thereby arguing for convergent evolution[47]. Gram-positive bacteria can accumulate polyphosphates but do not encode for polyphosphatekinase homologs as found in gram-negative bacteria. The feature of polyphosphate synthesis may have been acquired during phylogenesis to avail bacteria of subverting host surveillance. Bacterial polyphosphate chains interfere with various myeloid cell functions at the expense of the host. In the arms race against microbes, neutralization of polyphosphates may benefit from further exploration and optimization. Therapeutic polyphosphate targeting could provide an alternative or adjunctive treatment approach to help fight bacterial infections and their inflammatory complications.

## Methods

**Research animals.** All studies with mice were approved by the State Investigation Office of Rhineland-Palatinate, the Institutional Animal Care and Use Committee (IACUC) of Boston University and were in accordance with the guidelines of the U.S. National Institutes of Health, the German Animal Protection Act, and the Federation of European Laboratory Animal Science Associations, directive 2010/63/EU of the European Parliament and of the Council of the European Union. All mice were housed in a 45–60% humidity, 22 ± 2 °C ambient temperature, controlled light/dark (12 h/12 h) cycle, with free access to food and water and were used as sex−age-matched cohorts for experiments. C57BL/6J mice (8−12-week-old), Stat1$^{-/-}$ mice[48], Ifnar1$^{-/-}$ mice[49], Ifnβ$^{-/-}$ mice, P2y1$^{-/-}$ mice (all on C57BL/6J background) and Rage$^{-/-}$ mice (on A/J background) were maintained in

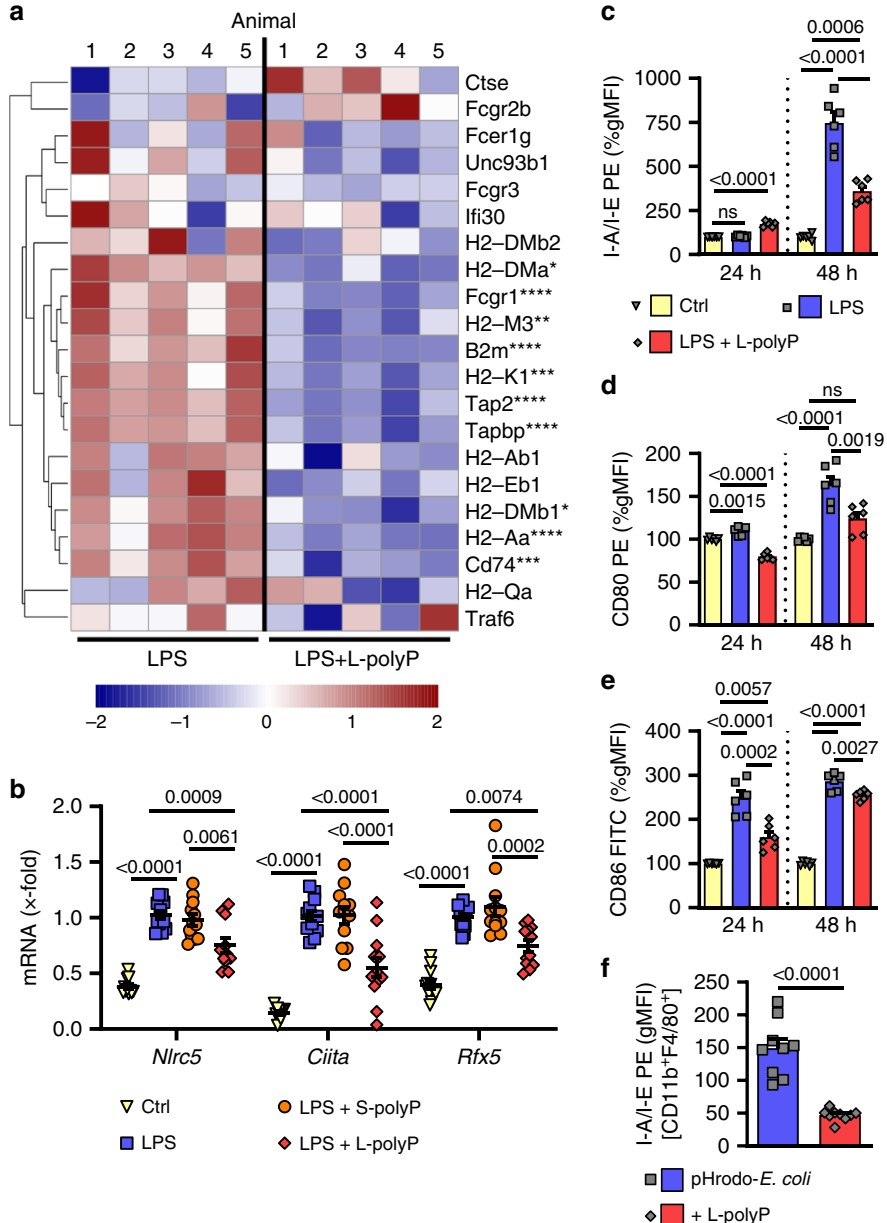

**Fig. 5 Interference of polyphosphates with MHCII antigen presentation. a** Heatmap of normalized expression (Z-score from −2/blue to +2/red) of genes associated with antigen processing and presentation of exogenous peptide antigen (GO:0002478) comparing LPS alone to its combination with long-chain polyphosphates (L-polyP), $n = 5$/group, RNA-seq. Statistical significances of gene expressions are marked by asterisks (displaying the adjusted $p$ value after DESeq2 analysis). **b** RT-qPCR of MHCI/II associated transcription factors *Nlrc5*, *Ciita*, and *Rfx5* in macrophages incubated with LPS ± S-/L-polyP, 12 h. **c–e** Normalized gMFI of MHCII molecules I-A/I-E (**c**) and costimulatory proteins CD80 (**d**) and CD86 (**e**) on CD11b+F4/80+ macrophages after LPS ± L-polyP, 24 and 48 h, flow cytometry. **f** MHCII (I-A/I-E) surface expression on CD11b+F4/80+ peritoneal macrophages 24 h after intraperitoneal pHrodo-*E. coli* ± L-polyP, flow cytometry ($n = 9$ mice/group). Data are expressed as mean ± s.e.m. **a** Wald test, **b–e** one-way ANOVA, **f** two-sided $t$ test, $n = 10−12$ samples/group (**b**) or $n = 6$ samples/group (**c–e**) examined over three independent experiments, **$p < 0.01$, ***$p < 0.001$, ****$p < 0.0001$, ns not significant ($p > 0.05$).

a specific pathogen-free environment. Germ-free and monocolonized Swiss Webster mice were housed in sterile flexible film isolators[50,51].

**Escherichia coli peritoneal infection.** *Escherichia coli* strains (MG1655 and Δ*ppk*/MJG224) were described and characterized before[14]. An overnight culture of bacteria was inoculated into LB medium (Luria/Miller from Carl Roth, Karlsruhe, Germany) and cultured at 37 °C with gentle shaking to an optical density at wavelength 600 nm of 0.5−0.6 (Nanodrop 2000c, Thermo Fisher Scientific, Waltham, MA, USA) to achieve a logarithmic growth phase. Bacteria were pelleted at 2500 × $g$ for 10 min and washed with phosphate buffered saline (PBS) (without Ca$^{2+}$ or Mg$^{2+}$ from Thermo Fisher Scientific). C57BL/6J male mice of 8–12 weeks of age received an intraperitoneal injection of 100 µl PBS with 0.8–1.6 × 10$^9$ *E. coli* wild type (WT; MG1655). The mice received one additional intraperitoneal injection with either inorganic polyphosphates

in a dose of 10 µg/g body weight (BW) for S-polyphosphates or L-polyphosphates, a dose of 2 µg/g BW recombinant exopolyphosphatase (Ppx) from *Saccharomyces cerevisiae*, or an equivalent volume of sterile PBS. To verify the infectious dose, the remaining washed *E. coli* were streaked on sheep blood agar plates (BD Bioscience, Franklin Lakes, NJ) to determine the exact numbers of CFU after 24 h incubation at 37 °C. The survival of *E. coli* challenged mice was monitored for at least 7 days. For nonsurvival experiments, the mice were euthanized at the indicated time points followed by peritoneal lavage infusing sterile PBS and determination of CFU on sheep blood agar plates as described above.

**CLP sepsis in monocolonized mice.** Sex- and age-matched germ-free Swiss Webster mice received 200 µl of *E. coli* WT or Δ*ppk* overnight LB culture via gavage for monocolonization as described before[51]. The intestinal colonization was

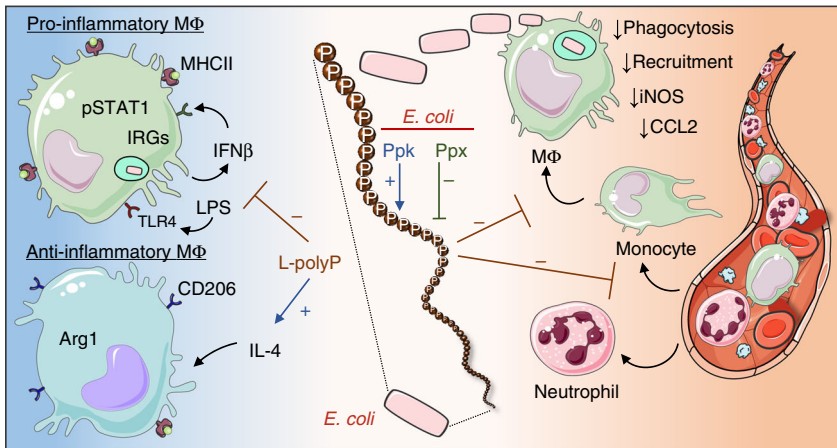

**Fig. 6 Schematic of polyphosphate effects on myeloid cells.** Long-chain polyphosphate (L-polyP) generation in *E. coli* is balanced by polyphosphatekinase (Ppk) and exopolyphosphatase (Ppx). Polyphosphates inhibit (−) the local recruitment of monocytes to the site of infection and their maturation into bactericidal macrophages (MΦ) including suppression of inducible nitric oxide synthase (iNOS). The polyphosphates further reduce an effective clearance of bacteria by counteracting phagocytosis in macrophages, monocytes and neutrophils. The LPS/TLR4-induced polarization of macrophages into the proinflammatory type ("M1") is opposed by polyphosphates. In addition, polyphosphates antagonize the production and response to IFNβ, interferon-regulated genes (IRGs), MHCII and STAT1 signaling. Simultaneously, long-chain polyphosphates enhance (+) the expression of some anti-inflammatory markers ("M2").

monitored by quantification of 16S rDNA from feces samples as published before[51]. After 14 days, the mice were removed from sterile isolator conditions to surgically induce sepsis by the cecal ligation and puncture (CLP) procedure[52]. In brief, the disinfected abdomen of anesthetized mice was opened with a 1 cm mid-level incision. The distal part (1 cm) of the cecum was ligated and punctured once with a 23G needle (Becton Dickinson, Franklin Lakes, NJ). The abdominal wall was closed using surgical sutures and mice received 1 ml of sterile 0.9% NaCl (B. Braun, Melsungen, Germany) by subcutaneous injection into the nuchal fold. Survival was monitored for 7 days, or mice were sacrificed after 48 h and peritoneal inflammatory cells were lavaged using 5 ml of sterile PBS + 0.5 mM ethylenediaminetetraacetic acid (EDTA (Promega, Fitchburg, WI, USA)). Colony-forming units were determined as described above. The peritoneal lavage cells were pelleted by centrifugation ($300 \times g$, 5 min, 4 °C) for subsequent flow cytometry studies and supernatants were stored at −80 °C until further analysis.

**In vivo cell migration assay**. Mice received a peritoneal injection of 50 μl/g BW 2.4% (w/v) thioglycolate (BD Bioscience) and additional 10 μg/g BW S-/L-polyphosphates or an equivalent volume of sterile PBS. At the end of experiments, mice were euthanized followed by lavage of peritoneal cells and flow cytometry-based cell counting.

**Macrophage preparation and culture**. Bone marrow-derived macrophages (BMDM) were generated by culturing bone marrow cells with L929 cell-conditioned medium for 7 days[53,54]. Peritoneal elicited macrophages were isolated 4 days after i.p. injection of 1.5 ml 2.4% (w/v) thioglycolate. In vitro cultivation of macrophages was performed in RPMI 1640 (Thermo Fisher Scientific) supplemented with 100 U/ml penicillin−streptomycin (Thermo Fisher Scientific) and 0.1% (w/v) bovine serum albumin (Carl Roth, Karlsruhe, Germany) at 37 °C, 5% $CO_2$, and 95% humidity.

**Flow cytometry**. The cells were washed in ice-cold sterile PBS ($300 \times g$, 5 min, 4 °C) and stained for 30 min with fixable viability dye eFluor 780 (Thermo Fisher Scientific, 1:1000) using heat-killed (1 min, 65 °C) cells as positive controls. Next, cells were washed twice with FACS buffer (0.25% (w/v) BSA, 0.02% (w/v) sodium azide, 2 mM EDTA in sterile PBS), preincubated for 15 min with anti-CD16/CD32 Fc-block antibody (10 μg/ml; BioLegend, San Diego, CA, 1:50) in FACS buffer followed by 30-min incubation with fluorescence dye-conjugated, anti-mouse antibodies for CD11b (clone M1/70, 1:100), Ly6G (clone 1A8, 1:50), Ly6C (clone HK1.4, 1:80), F4/80 (clone BM8, 1:100), I-A/I-E (clone M5/114.15.2, 1:80), CD80 (clone 16-10A1, 1:40), CD86 (clone GL-1, 1:50) and corresponding isotype controls (all from BioLegend).

For polyphosphate binding studies, biotinylated L-polyphosphates (50 μM) were incubated with macrophages on ice in the presence of phosphatase inhibitor cocktail 2 (Sigma-Aldrich) followed by 15 min with Streptavidin-PE (#405203, BioLegend, 1:50), washing and fixation with 4% PFA.

For intracellular staining, the cells were fixed/permeabilized with Cytofix/Cytoperm (BD Bioscience) for 20 min followed by washing with Perm/Wash buffer (BD Bioscience) and incubation with antibodies for CD206 (clone C068C2, BioLegend, 1:80) or iNOS (clone CXNFT, Thermo Fisher Scientific, 1:1000) and

corresponding isotype controls for 30 min on ice. Samples were washed twice with Perm/Wash buffer and re-suspended in FACS buffer.

For phospho-flow cytometry, cells were washed with FACS buffer after stimulation, fixed for 20 min with Cytofix (BD Bioscience), washed with FACS buffer, re-suspended in pre-cooled (−20 °C) Perm III buffer (BD Bioscience) and incubated overnight at −20 °C. Thereafter, samples were washed again with FACS buffer, incubated for 15 min on ice with anti-CD16/CD32 blocking antibody (BioLegend) before antibodies against phospho-STAT1[Y701] (clone 4a, BD Bioscience, 1:10), CD11b and F4/80 (1:200 each) were added for additional 30 min. Cells were washed with Perm/Wash and re-suspended in FACS buffer.

At least 50,000 events of interest were acquired on a FACSCanto II (BD Bioscience). For cell counting, 123count eBeads (Thermo Fisher Scientific, 1:5) were added according to the manufacturer's instructions. FlowJo VX Software (FlowJo, Ashland, OR) was used for data analysis.

**Phagocytosis assays**. Mice received an injection of 100 μg pHrodo™ green-*E. coli* (Thermo Fisher Scientific) in the presence or absence of 10 μg/g BW L-polyphosphates. Small volumes of blood were sampled in EDTA-tubes from the retro-orbital sinus of anaesthetized mice at several time points. Mice were sacrificed after 4−24 h and peritoneal cells were collected as described above and processed on ice. Following flow cytometry surface marker staining, live cells were analyzed using a FACSCanto II (Becton Dickinson) as described above. To control for auto-fluorescence, additional mice were challenged with fixed *E. coli* without pHrodo™ conjugate and used to obtain fluorescence minus one (FMO) controls. To determine the amounts of total undigested pHrodo™ green-*E. coli* bioparticles, 200 μl of peritoneal lavage was centrifuged 5 min at $10,000 \times g$, the pellet was re-suspended and incubated for 10 min in 2 N $H_2SO_4$ and pHrodo™ green fluorescence was determined with a Fluoroskan Ascent FL (Thermo Fisher Scientific). For the in vitro phagocytosis assay, BMDM were challenged with 330 μg/ml pHrodo™ green-*E. coli* in the presence or absence of 50 μM S- or L-polyphosphates. The fluorescence signals were acquired with a Fluoroskan Ascent FL at 4 h after stimulation.

**Live cell fluorescence imaging**. BMDM ($5 \times 10^4$/well) were seeded in 8-well chamber slides (Thermo Fisher Scientific) in RPMI 1640 medium without phenol red, supplemented with 100 U/ml penicillin-streptomycin (Thermo Fisher Scientific) and 0.1% (w/v) bovine serum albumin (Carl Roth) and allowed to adhere firmly overnight. The next day, BMDM were washed and maintained in HBSS for the staining procedure. After treating the surfaces with biotin/streptavidin blocking reagent (Avidin/Biotin Blocking Kit, Thermo Fisher), preformed complexes of 50 μM biotin-labeled L-polyphosphates and Streptavidin-PE or Streptavidin-FITC (BioLegend), generated by 30-min incubation of the reagents on ice, were added to the cells for 1 h at 37 °C. LysoTracker Green DND-26 (100 nM, Thermo Fisher Scientific) or CellMask Orange Plasma Membrane Stain (5 μg/ml, Thermo Fisher Scientific) were added gently together with the nuclear dye, Hoechst 33342 (Thermo Fisher Scientific), for another 30 min of incubation. The macrophages were washed thoroughly with HBSS to remove excess dyes and immediately imaged using a Zeiss LSM 710 NLO confocal laser scanning microscope (CLSM) with a 1.4 Oil DIC M27 ×63 plan-apochromat oil objective.

**Enzyme-linked immunosorbent assay (ELISA)**. ELISA kits for CXCL10 (IP-10) and CCL2 (MCP-1) were purchased from R&D Systems (Minneapolis, MN, USA), and the IFNβ ELISA was from Pestka Biomedical Laboratories (Piscataway, NJ). Cell-free supernatants and lavage fluids, or plasma samples were analyzed following the manufacturer's instructions and optical densities measured on an Opsys MR Dynex microplate reader (Dynex Technologies)[55].

**Western blotting**. The supernatants were carefully discarded and 10 cm dishes with attached macrophages were snap-frozen in liquid nitrogen and stored at −80 °C until further analysis. For cell lysis, 300 µl RIPA buffer (Merck Millipore, Billerica, MA, USA) supplemented with inhibitors for proteases (cOmplete protease inhibitor, Hoffmann-La Roche, Basel, Switzerland) and phosphatases (phosphatase inhibitor cocktail 2, Sigma-Aldrich) were added. Following 30-min incubation on ice, the cell debris was pelleted by centrifugation (10,000 × $g$, 10 min, 4 °C). Protein concentrations were determined at 595 nm using Bradford Ultra™ (Expedeon, Over Cambridgeshire, United Kingdom) and an Opsys MR microplate reader. Protein samples (20 µg per lane) were mixed with Laemmli buffer[56] and heated for 7 min to 95 °C. Gel electrophoresis (Bio-Rad, Hercules, CA) was performed and proteins transferred to an activated polyvinylidene difluoride membrane (Hoffmann-La Roche). Membranes were blocked for 1 h in TBS supplemented with 0.1% (v/v) Tween 20 (Sigma-Aldrich) and 5% (w/v) skim milk powder (AppliChem, Darmstadt, Germany) at room temperature followed by primary antibody application at 1:1000 in blocking buffer overnight at 4 °C. Primary antibodies for α-actinin (Cat. #3134), STAT1 (Cat. #9172 S) and phospho-STAT1$^{Y701}$ (Cat. #9167S) were from Cell Signaling Technology (Danvers, MA, USA). After three washing steps with TBS + 0.1% (v/v) Tween 20 (TBST) for 10 min each, membranes were incubated 90 min at room temperature with secondary antibody (horseradish peroxidase-conjugated anti-rabbit IgG1 from Vector Laboratories, Burlingame, CA, USA) as 1:5000 dilution in blocking buffer. Membranes were washed twice with TBST and once with TBS. Signal detection was performed using LumiGlo (Cell Signaling) and a Fusion FX (Vilber Lamourt, Collégien, France). Signal quantification was accomplished by built-in densitometry analysis of the Fusion FX software (V16.11). Ratios of phospho-STAT1$^{Y701}$ divided by STAT1 were calculated and normalized to values of the LPS stimulation.

**RNA-seq**. BMDMs (C57BL/6) were differentiated and stimulated as described above. Total RNA was isolated using the Qiagen RNeasy kit with an additional on-column DNA digestion step (Qiagen). RNA integrity was evaluated with an RNA 6000 Nano total RNA kit (Agilent Genomics, Santa Clara, CA, USA) on a 2100 Bioanalyzer (Agilent Genomics). Library preparation was performed following the manufacturer's instructions (TruSeq stranded mRNA kit, Illumina, San Diego, CA, USA) and samples were sequenced as 67-nt single-end reads with a total of two runs (Hiseq Rapid SBS kit v2 on an Illumina HiSeq 2500, Illumina). Library preparation and sequencing was performed under the supervision of Dr. C. Han at the Genomics Core Facility of the Institute of Molecular Biology (University of Mainz, Germany). Raw reads per sample ranged from $22.7-31.7 \times 10^6$. In STAR aligner (v2.4.0b), ~98% of the reads could be mapped to the index mouse genome GRCm38 from ENSEMBL (annotation ENSEMBL v76[57]) with ~90% unique mapping rates. Quantification was performed using featureCounts from RSubread (v1.26.1). Exploratory data analysis was performed with the pcaExplorer package (v2.2.1)[58]. Differential expression analysis was performed with the ideal package (v1.0.0)[59]. DEGs were considered significant if p-adj (adjusted $p$ value) < 0.05 from DESeq2 (v1.16.0). Further analyses included Gene Ontology[60] pathway enrichment by goseq (v1.28.9) and topGO (v2.28.0) (setting all mapped genes as background dataset) and matching DEGs against the *Interferome v2.0* database[31].

**RT-qPCR**. Total RNA was isolated from cells using the Qiagen RNeasy mini kit (Qiagen, Venlo, Netherlands). Reverse transcription of cDNA was performed with the High-Capacity cDNA Reverse Transcription kit (Life Technologies, Carlsbad, CA) according to the manufacturer's instructions using a Mastercycler pro S (Eppendorf, Hamburg, Germany). Quantitative PCR was performed on a C1000 with CFX real-time PCR detection system (Bio-Rad Laboratories) using iQ SYBR green mix (Bio-Rad Laboratories). Target gene expression was compared between samples by normalizing to *Gaph* expression and applying the $2^{-\Delta\Delta Ct}$ formula[61]. The primer sequences are listed in Supplementary Table 1.

**Production of recombinant exopolyphosphatase (Ppx)**. The recombinant His-tagged polyphosphate-degrading exopolyphosphatase from *Saccharomyces cerevisiae* (Ppx) was expressed in *E. coli* (pScPpx2) using a plasmid with Ampicillin resistance cassette[14,62]. Overnight cultures of this strain were prepared in LB medium plus Ampicillin (100 µg/ml; Sigma-Aldrich). Isopropyl $\beta$-D-1-thiogalactopyranoside (IPTG; 1 mM; Thermo Scientific) and fresh Ampicillin were added followed by shaking incubation for 4 h at 37 °C. Bacteria were harvested by centrifugation, lysed in sodium phosphate buffer supplemented with lysozyme (1 mg/ml; Sigma-Aldrich), benzonase endonuclease (50U/ml; Sigma-Aldrich) and MgCl$_2$ (2 mM; Carl Roth, Karlsruhe) followed by sonication (Q700, QSonica), centrifugation and filtration. Next, the supernatants containing histidine-tagged recombinant Ppx protein were passed through nickel ion affinity columns (HisTrap FF Crude 1 ml, GE Healthcare) by FPL chromatography (ÄktaPure, GE

Healthcare), dialyzed against TrisHCl/KCl buffer (ph = 7.5) supplemented with 10% glycerol and were stored at 4 °C. Pre- and post-elution fractions were controlled/validated by protein gel electrophoresis (Ppx: ~45 kDa). The specific enzymatic activity of Ppx was confirmed to fall in the range of $500-800$ nmol P$_i$/min mg/Ppx by polyphosphate digestion followed by colorimetric monophosphate quantification (BIOMOL Green assay, Enzo Life Sciences).

**Detection of polyphosphates**. For quantification of polyphosphates by gel electrophoresis, S-/L-polyphosphates were adjusted to 1 mM monophosphate concentration with or without digestion using 100 U/ml of calf intestinal alkaline phosphatase (CIP, New England BioLabs, Ipswich, MA) overnight at 37 °C. Five micrograms of undigested and digested S-polyphosphate and L-polyphosphate were separated with an urea polyacrylamide gel followed by positive staining with 4′,6-diamidino-2-phenylindole (DAPI, Sigma-Aldrich) as described before[63]. Fluorescent images were acquired using a Gel Doc EZ Imager (Bio-Rad Laboratories, Hercules, CA).

Polyphosphates in peritoneal lavage fluids were quantified using a MicroMolar Polyphosphate Assay Kit (ProFoldin, Hudson, MA) according to the manufacturer's instructions. Briefly, 60 µl of lavage samples were mixed with 60 µl of polyphosphate-specific dye and incubated for 5 min. L-polyphosphates were used to generate a standard curve. The light emission (550 nm) of samples and standards after excitation at 415 nm were measured on a SpectraMax i3 (Molecular Devices, Sunnyvale, CA). This assay detected 1.1 ± 0.7 µM (S.E.M.) polyphosphates in peritoneal lavage fluids (10 ml/mouse) of untreated healthy C57BL/6 J mice (sham).

**Cell viability assay**. Macrophages were cultured in RPMI 1640 medium without phenol red (Thermo Fisher Scientific) supplemented with 100 U/ml penicillin-streptomycin and 0.1% (w/v) BSA with S-/L-polyphosphates (50 µM) for 24 h. To evaluate cell viability, the lactate dehydrogenase (LDH) activity in supernatants was measured using a colorimetric CytoTox 96® assay (Promega) according to the manufacturer's instructions. As positive controls, untreated resting macrophages were disrupted in lysis buffer and used along with samples of supernatants. After 30 min, the substrate reaction was stopped and formazan generation measured at a wavelength of 490 nm (Opsys MR Microplate Reader).

**Griess assay**. Nitrite concentrations as a surrogate for nitric oxide formation were determined in cell culture supernatants using the Griess assay (Thermo Fisher Scientific) with minor adjustments to the manufacturer's instructions. Briefly, 75 µl of supernatants or serial dilutions of a nitrite standard in RPMI 1640 medium were mixed with 75 µl Griess reagent and incubated for 30 min at room temperature. The colorimetric signals were detected at a wavelength of 550 nm (Opsys MR Microplate Reader). Nitrates originating from the use of RPMI medium did not lead to a detectable signal using this method, as proven by a blank test reaction performed with RPMI.

**Reagents**. If not stated otherwise, the following concentrations of reagents were used: 100 ng/ml lipopolysaccharide (LPS; *E. coli* O111:B4, Sigma-Aldrich, St. Louis, MO), 10 µg/ml Zymosan (ZYM; *S. cerevisiae*, InvivoGen, San Diego, CA), 10 µg/ml Peptidoglycan (PGN; *E. coli* O111:B4, InvivoGen), 10 µg/ml Flagellin (FLA; *B. subtilis*, InvivoGen), 1 µg/ml dsDNA (*E. coli* K12, InvivoGen) transfected using Oligofectamine, 500 U/ml recombinant mouse IFNβ (Pestka Biomedical Laboratories), 10 ng/ml recombinant mouse IL-4 (Peprotech), and 100 µg/ml dextran sulfate (Sigma-Aldrich). S-polyphosphates had a modal length of 70 phosphates (range: 25–125 phosphates) and L-polyphosphates (unmodified or biotinylated) had a modal length of 700 phosphates (range: 200–1300 phosphates) and were solubilized from high-molecular-weight polyphosphate as described before[64]. S-/L-polyphosphates were used as 50 µM (indicated as concentration of monophosphates) unless stated otherwise in the figures.

**Statistics and data presentation**. GraphPad Prism 8.0 software was used to analyze and display data. The sample sizes to derive statistics and numbers of biologically independent experiments are provided in the figure legends. Data in bar graphs are depicted as mean ± standard error of the mean (s.e.m.). The two-sided Student's $t$ test or nonparametric Mann−Whitney test were applied to compare two groups and ANOVA (one-way, two-way) was used for comparisons of multiple groups or time points. Survival studies were analyzed using two-sided Log-rank (Mantel−Cox) and Gehan−Breslow−Wilcoxon tests. Statistical analyses of RNA-seq data sets were performed using DESeq2 (v1.16.0) and the elim method of topGO (v2.28.0)/goseq (v1.28.0) for Gene Ontology pathway enrichment analysis[65]. Statistical $p$ values were corrected for multiple testing by using the FDR (false discovery rate) controlling method by Benjamini and Hochberg. Significance was considered for $p \leq 0.05$. *$p \leq 0.05$, **$p \leq 0.01$, ***$p \leq 0.001$, and ****$p \leq 0.0001$, ns not significant ($p > 0.05$).

Figure 1a was created using the ChemSketch v14 software from ACD/Labs. Figure 6 was created with elements from Servier Medical Art by Servier under a Creative Commons Attribution 3.0 Unported License. Supplementary Fig. 1e attributes a modified element to the free icons library (https://icon-library.com/icon/mice-icon-10.html; Mice Icon #136773). The featured image combines Fig. 1a, e, f.

**Reporting summary**. Further information on research design is available in the Nature Research Reporting Summary linked to this article.

## Data availability

The RNA-seq data presented in this manuscript have been deposited in the Gene Expression Omnibus (GEO) under accession number GSE131561. All other data are available from the corresponding author upon reasonable request. Source data are provided with this paper.

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

## Acknowledgements

We cordially thank Foruzandeh Samangan for technical assistance and Erica Cadigan for secretarial assistance. We thank Dr. Ursula Jakob and Dr. Michael Gray for the *E. coli* strains and helpful comments. We thank Dr. Thomas Staffel (BK Giulini) for providing starting material for S-polyphosphate preparations, Dr. Christian Gachet and Dr. Kristine Gampe for providing bone marrow of *P2Y1*$^{-/-}$ mice, and Dr. Dennis Strand for technical support with fluorescence imaging. M.B. thanks Dr. Jay Mizgerd for mentorship and reading the manuscript. This work was financed by the Federal Ministry of Education and Research (01EO1003, 01EO1503 to M.B., C.R. and F.M.), the Deutsche Forschungsgemeinschaft (BO3482/3-3, BO3482/4-1 to M.B., RE-3450/5-2 to C.R.), National Institutes of Health (1R01HL141513, 1R01HL139641 to M.B. and R35HL135823 to J.H.M.), a Marie Curie Career Integration Grant of the European Union (Project 334486 to M.B.), a Clinical Research Fellowship of the European Hematology Association (to M.B.), a project grant from the Boehringer Ingelheim Foundation (Consortium Grant "Novel and neglected cardiovascular risk factors") to C.R., and the Austrian Science Fund (FWF; SFB-F6101 and SFB-F6106 to M.M. and B.S.). C.R. was awarded a fellowship from the Gutenberg Research College. The authors are responsible for the content of this publication. Some data from this paper are used by G.S. for his Doctor of Medicine thesis.

## Author contributions

J.R. designed and performed experiments, analyzed data and wrote the manuscript, G.S., M.S. and A.S. designed and performed experiments and analyzed data, F.M. analyzed RNA-seq data and provided support for statistical analyses, A.M. performed immuno-blotting experiments, B.S. and M.M. provided *Ifnar1*$^{-/-}$, *Stat1*$^{-/-}$ and *Ifnβ*$^{-/-}$ cells and helpful comments, Z.K. provided *Rage*$^{-/-}$ tissues and helpful comments, C.R. performed, supported and supervised experiments with germ-free and monocolonized mice, S.A.S. and J.H.M. provided polyphosphates, protocols and helpful comments, M.B. conceived and supervised the study, designed experiments and interpreted data, provided funding and wrote the paper.

## Competing interests

J.H.M. and S.A.S. are inventors on patents and patent applications on medical uses of polyphosphates and polyphosphate inhibitors, and the laboratory of J.H.M. receives support from sales of polyphosphates and related reagents through Kerafast, Inc. The other authors declare no competing interests.
