## [Peer Review File · Nature Communications]

Reviewers' comments:

Reviewer #1 (Remarks to the Author):

In this manuscript, Roewe, et al. investigate the relationship between bacterial polyphosphate (polyP) and the mammalian immune response following infection. In particular study addresses the important question of how bacteria might be able to counteract the antimicrobial response via production of long polyP chains.

The work begins with phenotypic analyses using a sepsis model. The authors demonstrate the ability of long-polyphosphates, characteristic of those made by bacteria, to impair survival of mice by decreasing myeloid cell response post-sepsis. The reverse also appears to be true -mice infected with *E. coli* that can't make any polyP (Δ ppk) show increased survival. The number of myeloid cells and their associated immune by-products (such as iNOS) also are also impacted by exogenous long-chain polyP and by the ppk status of the *E. coli* used for infection.

The second part of the work involves an RNA-seq analysis of the response to LPS treatment of macrophages in the presence or absence of exogenous long-chain polyP. The authors use this analysis as a staging point to describe long-chain polyP-mediated inhibition of M1 macrophage gene expression and an increase in M2 gene expression, antagonism of type I interferon response and decreased MHCII-mediated antigen presentation.

The overall model seems to be that bacteria make long-chain polyphosphates during infection and that these polyP chains interfere with various myeloid cell functions to promote *E. coli* survival at the expense of the host.

Overall this is a nice study. The work is generally well controlled, the experiments convincing, and conclusions solid. The manuscript is descriptive rather than pinpointing a specific mechanism. However, more than anything else, this reflect the state of the field and the inherent difficulties in studying polyphosphates in vivo. Despite the lack of mechanism, I do feel that the work represents an important contribution. However, there a number of concerns I would like to see addressed before the manuscript is suitable for publication.

1. The authors find polyphosphate in the peritoneal cavity after infection (extended data 1g). How does this compare with uninfected mice, and are the chain lengths representative of those presumed to be found in bacteria?

Is the model that bacterial polyphosphate is interacting with receptors on the outside of the myeloid cells? If so, how do the authors suppose that the polyP is released from the bacteria?

2. The authors should clarify in the main text if polyP concentrations are measured in terms of free phosphate units.

3. Figure 1:

a) There seem to be some inconsistencies in Fig 1 with regards to the survival in response to wild-type polyP. Can the authors comment on this?

b) Figure 1B should ideally have a control where long-chain polyP is assayed along-side of the short chain.

c) Can the authors comment further on the drastic impact of PPX (survival 100 %) – Why should this effect be any greater than that of the Δppx mutant strain?

4. RNA-seq analyses:

a) Are 'significant' genes, GO-terms and other analyses corrected for multiple testing?

b) Figure 2a (and other figures similar): Why are the non-significant genes being shown here? What would the trends look like if they just showed the significant hits?

c) Did the genes Arg1, Ym1, Fizz1 appear 'regulated' in the RNA-seq?

5. Article format seems too condensed: The relevance and impact of the paper could be made clearer by expanding the introduction and discussion. As an example, the work of Terashima-Hasegawa et al. (ref 21) and how this relates to the current study seems to have been glossed over in the discussion. In the results section, subheadings and further details regarding the rationale for each experiment, and how these details fit into the context of a broader model are warranted.

6. If space permits, the authors should provide a model for how everything fits together, including a discussion of the unknowns (see above), could be provided as an additional figure.

7. Significance lines and asterisks are not always clear in figures. Some lines do not have their own asterisks. For example, for Figure 4B NLRC5, is the difference between short and long polyP **, NS, or something else.

8. Why are there lines on the WB in Figure 3H? Were the control, LPS and LPS+polyP samples run on the same gel, for each antibody used? It would be inaccurate to compare treatment groups if samples were not analyzed on the same blots for each antibody. (I think it would be fine to do the total and phosho antibodies on separate blots, however).

9. The authors could provide additional references to support their assertion that polyP chains made in human cells are 'short'. Indeed, the Kornberg lab has previously seen longer chains in mammalian tissues (PMID:7890711).

Reviewer #2 (Remarks to the Author):

Production of polyphosphate polymers is a ubiquitous trait of bacteria. In this study, Roewe J and co-workers describe a role for long polyphosphates in immune suppression by bacteria. They show that long polyphosphates either produced by *E. coli* or exogenously co-administered into mice impede LPS-mediated inflammation and bacterial clearance. Overall the manuscript is clearly written and well controlled. This is a good study and well should be published in Nature Communications swiftly. I only have a few suggestions for improvement.

Main points

1) The authors have mainly focussed on TLR4-IFN-I signaling. However it seems that the inhibitory effect of polyphosphates is global. Do polyphosphate polymers also block other innate immune receptor signaling pathways relevant for anti-bacterial immunity for example TLR2 and TLR9 mediated responses.

2) One caveat of the study is the lack of defined mechanisms. Whereas the observed effects are likely due to overlapping mechanisms as discussed by the authors, can the authors to test whether polyphosphates polymers acts by targeting fundamental cellular processes important for receptor signaling, for example phagocytosis/endocytosis.

3) Upon adding on to cells, are polyphosphates internalized? Or do they interact with and block immune agonists?

Nelson O. Gekara

Stockholm University

**Revised Manuscript for Consideration
NCOMMS-19-18094**

Title: “*Bacterial Polyphosphates Interfere with the Innate Host Defense to Infection*”

Point-to-point responses to the Reviewers' comments:

Reviewer #1 (Remarks to the Author):

In this manuscript, Roewe, et al. investigate the relationship between bacterial polyphosphate (polyP) and the mammalian immune response following infection. In particular study addresses the important question of how bacteria might be able to counteract the antimicrobial response via production of long polyP chains.

The work begins with phenotypic analyses using a sepsis model. The authors demonstrate the ability of long-polyphosphates, characteristic of those made by bacteria, to impair survival of mice by decreasing myeloid cell response post-sepsis. The reverse also appears to be true -mice infected with E. coli that can't make any polyP (Δ ppk) show increased survival. The number of myeloid cells and their associated immune by-products (such as iNOS) also are also impacted by exogenous long-chain polyP and by the ppk status of the E. coli used for infection.

The second part of the work involves an RNA-seq analysis of the response to LPS treatment of macrophages in the presence or absence of exogenous long-chain polyP. The authors use this analysis as a staging point to describe long-chain polyP-mediated inhibition of M1 macrophage gene expression and an increase in M2 gene expression, antagonism of type I interferon response and decreased MHCII-mediated antigen presentation.

The overall model seems to be that bacteria make long-chain polyphosphates during infection and that these polyP chains interfere with various myeloid cell functions to promote E. coli survival at the expense of the host.

Overall this is a nice study. The work is generally well controlled, the experiments convincing, and conclusions solid. The manuscript is descriptive rather than pinpointing a specific mechanism. However, more than anything else, this reflect the state of the field and the inherent difficulties in studying polyphosphates in vivo. Despite the lack of mechanism, I do feel that the work represents an important contribution. However, there a number of concerns I would like to see addressed before the manuscript is suitable for publication.

We thank the reviewer for the positive comments. We have performed additional experiments to address the raised concerns. In addition, the discussion section was substantially expanded as suggested by the editor.

1. The authors find polyphosphate in the peritoneal cavity after infection (extended data 1g). How does this compare with uninfected mice, and are the chain lengths representative of those presumed to be found in bacteria?

Response:

We thank the reviewer for this comment. As suggested, we have now determined the background levels of polyphosphates in healthy, uninfected mice (Supplementary Fig. 1g). The levels in

peritoneal lavage fluids ($1.1 \pm 0.7 \mu\text{M}$) were lower as in WT *E. coli* infected mice ($8.6 \pm 1.6 \mu\text{M}$) and Δppk *E. coli* infected mice ($3.0 \pm 1.6 \mu\text{M}$). The used polyphosphate assay does not discriminate between short and long polyphosphates (i.e. platelet-derived vs. bacteria-derived), but according to the manufacturer's specifications does not cross-react with monophosphates, diphosphates, triphosphates, ATP, GTP, ADP, GDP etc. Unfortunately, there are currently no commercial assays available to determine the chain length of polyphosphates. This fact reflects the infant state of the field. The method of gel electrophoresis and DAPI staining is not sufficiently sensitive for diluted peritoneal lavage samples. In a recent report, the WT *E. coli* PPK enzyme has been demonstrated to produce long-chain polyphosphates with molecular sizes of more than $>300 P_i$ residues (new Ref. 45). While *ppk* mutations could affect the amount of polyphosphate accumulation in *E. coli*, the variations in polyphosphate chain length were found to be only modest (Ref. 45).

We have added a paragraph on the current limitations of polyphosphates detection methods and chain length considerations in *E. coli* with new references to the discussion section.

Is the model that bacterial polyphosphate is interacting with receptors on the outside of the myeloid cells?

Response:

We have performed additional experiments using biotinylated long-chain polyphosphates as molecular probes. As shown in new Fig. 2a and 2b, polyphosphates avidly bind to the surface of macrophages (incubation for 30 min on ice to prevent internalization). The binding of polyphosphates is antagonized by polyanionic dextran sulfate, which neutralizes the positive charge of the exposed surface proteins on macrophages. This finding suggests that at least the initial interaction of polyphosphates with macrophages relies on electrostatic binding forces.

Furthermore, we now include data of *live cell* imaging of macrophages cultured at 37°C (new Fig. 2c). We found that polyphosphates were subsequently internalized and co-localized with intracellular membrane structures stained with CellMask Orange dye (new Fig. 2c).

It has been reported that short polyphosphates act through the RAGE and P2Y1 receptors (new References: 20, 21). These cited studies used RNA interference for silencing of RAGE and P2Y1 in human umbilical vein endothelial cells and rat astrocytes. Therefore, we have obtained macrophages derived from knockout mice for RAGE and P2Y1, but these cells remained equally responsive to bacterial polyphosphates (new Fig. 3f, 3g). This argues against RAGE and P2Y1 representing universal polyphosphate receptors. We rather believe that the overall polyphosphate effects may be a result of overlapping mechanisms from polyphosphates binding to hundreds of proteins in macrophages.

We have expanded the discussion section to address this point to the readers. To study the molecular interactions of polyphosphates with potential membrane-bound receptors is an important aim of our ongoing and future research efforts.

If so, how do the authors suppose that the polyP is released from the bacteria?

Response:

We thank the reviewer for this relevant question. Bacteria store polyphosphates in high concentrations in specialized organelles termed acidocalcisomes (100-200 nm size). It has also been reported that polyphosphates are externalized in association to the cell wall of some bacteria. While active release mechanisms of polyphosphates occur from the secretory granules of platelets and mast cells, no such mechanism has yet been described for bacteria. We believe that the dissociation of polyphosphates from bacteria is a passive process that takes place as a consequence of bacterial damage/injury and cell death. In the course of infection, bacteria are destroyed by different means of the immune defense (reactive oxygen species, toxic granules,

proteases) and are digested after phagocytosis in the phagolysosomes. We have added paragraphs regarding these important points to the introduction and discussion of the manuscript.

The authors should clarify in the main text if polyP concentrations are measured in terms of free phosphate units.

Response:

As suggested, we have clarified also in the main text that all numbers on polyphosphate concentrations are calculated as free monophosphate units.

3. Figure 1:

a) There seem to be some inconsistencies in Fig 1 with regards to the survival in response to wild-type polyP. Can the authors comment on this?

Response:

The differences of the survival rates of the control groups in Fig. 1b, 1c, 1e are explained by day-to-day variations. The *E. coli* infections are prone to slight variations in the daily infectious dose (page 15, line 326: *E. coli* $0.8-1.6 \times 10^9$). It is not feasible to obtain the exact same number of *E. coli* CFUs on each day, because of the low accuracy of real-time *E. coli* detection methods and the fast proliferation rates of *E. coli*. We used optical density measurements for *E. coli* numbers, that were confirmed by CFU counts on sheep blood agar plates the next day after infection. This is a standard approach used by many research groups. The Fig. 1f is a different sepsis model (CLP) as compared to live *E. coli* infections shown in Fig. 1b, 1c, 1e. We now comment on this in the manuscript for clarification.

b) Figure 1B should ideally have a control where long-chain polyP is assayed along-side of the short chain.

Response:

We totally agree with the reviewer that experiments with comparing 3 groups side-by-side (*E. coli* alone, S-PolyP, L-PolyP) would have been ideal. However, the studies were subject to quite strict constraints by the local ethics committee in Rhineland-Palatinate, Germany. At first, we could only obtain approval for using L-PolyP (Fig. 1c) before further approval for a set of experiments with S-PolyP (Fig. 1b) was granted. While we strongly disagree with the ethics approval process, we are legally bound to these regulations, which are also strict to prevent duplication of completed experiments.

c) Can the authors comment further on the drastic impact of PPX (survival 100 %) – Why should this effect be any greater than that of the Δppx mutant strain?

Response:

Fig. 1e shows survival rates of live *E. coli* i.p. injection improved by 36% (64% vs. 100%) with recombinant PPX treatment. In Fig. 1g, the survival rates after CLP sepsis of gnotobiotic mice, improved by 37% (38% vs. 75%), when monocolonized with polyphosphate-deficient Δppx *E. coli* (not Δppx).

These are two different experimental models and a direct comparison is difficult because of several variables and day-to-day variations as described above. We would like to also point out that there is no complete absence of bacteria-derived polyphosphates in the Δppx *E. coli* monocolonization group (Supplementary Data Fig. 1g), presumably due to a residual activity of other *ppk* isoforms. We have revised the manuscript by presenting a more detailed description in the main text to better clarify how the experiments were designed.

4. RNA-seq analyses:

a) Are 'significant' genes, GO-terms and other analyses corrected for multiple testing?

Response:

The significant differentially expressed genes are indeed corrected for multiple testing. We used the FDR (False Discovery Rate) controlling method by Benjamini and Hochberg.

The GO terms are not corrected for multiple testing for the following reasons:

1. The multiple testing adjustment procedures assume that the tests are independent from each other. However, gene sets are not independent from each other, given their intrinsic redundancy. Hence, it would break one assumption and deliver potentially misleading significance values.

2. We used the elim method of topGO. This method accounts for the GO topology, and computes the p-value of a GO term conditioned on the neighbouring terms. As describe above, the tests are therefore not independent and the multiple testing theory does not directly apply. According to the authors of the elim method from topGO, the p-values returned by these methods are interpreted as corrected or not affected by multiple testing. Please see the topGO vignette for further detail.

3. The ranking by significance was performed for giving an overview of what functional categories are mainly affected in the data sets (Supplementary Figures). We recommend using this analysis with caution. We mainly used it to design and perform further independent biological experiments, for example to test the hypothesis if polyphosphates interfere with the interferon response. We are aware that the importance of P-values for statistical reporting is subject of a current debate (PMID: 31314974).

The statistical RNA-seq analysis was lead by co-author Dr. F. Marini (Biostatistician). Additional information on the bioinformatics/biostatistics analysis and new references were added to the Methods section.

b) Figure 2a (and other figures similar): Why are the non-significant genes being shown here? What would the trends look like if they just showed the significant hits?

Response:

We apologize for any confusion. The MA plots show both significant and non-significant genes because we wanted to display the full M1/M2 polarization and IRG signatures, without being biased by a binary threshold like the significance cutoff (i.e. some genes would completely not be shown even if they had a p-value close to the <0.05 cutoff). In our experience, the significance values for gene expression after stimulation of macrophages (LPS, polyphosphates) are dynamic and depend on the time point of analysis. We have found that some genes with RNA-seq p-values close to the significance cutoff were clearly significant, when tested at earlier or later time points by RT-qPCR in independent biological replicates. In general, we have used the MA plots as a visualization, which depict at the same time effect size and average expression value. An alternative view could be a heatmap of these genes.

We have now included how the figures would look like with just the significant hits as requested (see *Review Materials* at the end of this letter). This alternative data presentation does not appear to change the primary conclusion that PolyP suppressed the expression of many M1 genes and increased the majority of M2 genes. However, this was not completely consistent for all genes suggesting that the PolyP effects on macrophage polarization were quite unique and not strictly according the M1/M2 paradigm.

c) Did the genes Arg1, Ym1, Fizz1 appear 'regulated' in the RNA-seq?

Response:

In fact, the M2-markers Arg, Ym1 and Fizz1 were not even expressed by BMDM (unstimulated, LPS or L-polyphosphates, 12h) in the RNA-seq data. As shown in Fig. 3h, these genes were strongly induced only during M2 polarization using recombinant IL-4 for 48 h, which was further modulated by L-polyphosphates.

5. Article format seems too condensed: The relevance and impact of the paper could be made clearer by expanding the introduction and discussion. As an example, the work of Terashima-Hasegawa et al. (ref 21) and how this relates to the current study seems to have been glossed over in the discussion. In the results section, subheadings and further details regarding the rationale for each experiment, and how these details fit into the context of a broader model are warranted.

We thank the reviewer for this reasonable comment and have added additional paragraphs to the introduction and discussion section. (The manuscript had been submitted under different formatting guidelines (i.e. strict word limit of 1500) with direct re-routing to Nature Communication (word limit: 5000) as per journal guidelines without a requirement for reformatting the manuscript length.)

6. If space permits, the authors should provide a model for how everything fits together, including a discussion of the unknowns (see above), could be provided as an additional figure.

Response:

We agree that this is a good idea and have added a schematic for visualization and clarification of the presented findings (new Fig. 6).

7. Significance lines and asterisks are not always clear in figures. Some lines do not have their own asterisks. For example, for Figure 4B NLRC5, is the difference between short and long polyP **, NS, or something else.

Response:

We have made the requested changes throughout the manuscript to avoid any confusion regarding marking of significances with asterisks. For Figure 4B NLRC5, the difference between short and long polyphosphates is ** $p < 0.01$.

8. Why are there lines on the WB in Figure 3H? Were the control, LPS and LPS+polyP samples run on the same gel, for each antibody used? It would be inaccurate to compare treatment groups if samples were not analyzed on the same blots for each antibody. (I think it would be fine to do the total and phospho antibodies on separate blots, however).

Response:

The samples were run on the same gels / blots for each antibody used. We used the lines merely for a more condensed and rearranged layout to save print space. We have included images of the uncropped original blots as new Supplementary Fig. 8.

9. The authors could provide additional references to support their assertion that polyP chains made in human cells are 'short'. Indeed, the Kornberg lab has previously seen longer chains in mammalian tissues (PMID:7890711).

Response:

We acknowledge the report by the Kornberg lab describing the existence of long-chain polyphosphates in the mammalian brain. We have expanded the list of reference and the discussion section for a more balanced view.

Reviewer #2 (Remarks to the Author):

Production of polyphosphate polymers is a ubiquitous trait of bacteria. In this study, Roewe J and co-workers describe a role for long polyphosphates in immune suppression by bacteria. They show that long polyphosphates either produced by *E. coli* or exogenously co-administered into mice impede LPS-mediated inflammation and bacterial clearance. Overall the manuscript is clearly written and well controlled. This is a good study and well should be published in Nature Communications swiftly. I only have a few suggestions for improvement.

Response:

We thank the reviewer for the positive comments.

Main points

1) The authors have mainly focussed on TLR4-IFN-I signaling. However it seems that the inhibitory effect of polyphosphates is global. Do polyphosphate polymers also block other innate immune receptor signaling pathways relevant for anti-bacterial immunity for example TLR2 and TLR9 mediated responses.

Response:

To improve the study, we have performed additional experiments using two different TLR2 agonists (Zymosan and Peptidoglycan), the TLR5 agonist Flagellin and double-stranded DNA as TLR9 agonist. As shown in new Figure 4f, polyphosphates also modulate CXCL10 release in macrophages, when the TLR2 and TLR9 pathways are activated. We thank the reviewer for this important suggestion.

2) One caveat of the study is the lack of defined mechanisms. Whereas the observed effects are likely due to overlapping mechanisms as discussed by the authors, can the authors to test whether polyphosphates polymers acts by targeting fundamental cellular processes important for receptor signaling, for example phagocytosis/endocytosis.

Response:

We completely agree that the search for defined mechanisms is of great importance for the ongoing and future research on polyphosphates. As shown in Fig. 1k, polyphosphates reduced the phagocytosis of pHrodo *E. coli* particles by macrophages, monocytes and neutrophils. We include new data, demonstrating that polyphosphates bind to the surface of macrophages (new Fig. 2a, 2b), which is antagonized by polyanionic dextran sulfate. This suggests that at least the initial interaction of polyphosphates with macrophages relies on electrostatic binding forces. Our studies with the dynasore endocytosis inhibitor were largely inconclusive (data not shown), presumably because many cellular functions are compromised by endocytosis inhibition.

It has been reported that short polyphosphates act through the RAGE and P2Y1 receptors (new references: 20, 21). These cited studies used siRNA knock-down of RAGE and P2Y1 in human umbilical vein endothelial cells and rat astrocytes. We have obtained macrophages derived from knockout mice for RAGE and P2Y1, but these cells remained equally responsive to bacterial polyphosphates (new Fig. 3f, 3g). These findings argue against RAGE and P2Y1 representing universal polyphosphate receptors. As acknowledged by the reviewer, the overall polyphosphate effects may be a result of overlapping mechanisms from polyphosphates binding to hundreds of proteins of macrophages. In our ongoing studies, we have performed affinity purifications followed by proteomics of these polyphosphate binding proteins, proteome microarray binding assays and generated novel knockout mouse lines for two candidate proteins (unpublished data). Unfortunately, so far these studies failed to reveal meaningful additional mechanistic insights with

relevance to the current manuscript. We have added text on the limitations of the current report to the discussion section.

3) Upon adding on to cells, are polyphosphates internalized? Or do they interact with and block immune agonists?

To answer the important question if polyphosphates are internalized, we have performed live cell fluorescence imaging studies at 37°C of macrophages, when incubated with biotin-labeled polyphosphates (new Fig. 2c). In fact, polysphosphates are internalized and co-localized with intracellular membrane structures stained with CellMask Orange dye (new Fig. 2c). So far, we could only find a weak co-localization with LysoTracker dye (data not shown).

Polyphosphates are polyanions and can interact with positively charged molecules. For example, polyphosphates bind to platelet factor 4 (CXCL4; new Reference: 41). However, a direct interaction of bacterial polyphosphates with TLR agonists or interferons seems unlikely to explain the observed effects because such effects are not seen with short polyphosphates (=control). We have expanded the discussion section of the manuscript to cover these important points for the readers in further detail.

Materials for Review Process

Bacterial Polyphosphates Interfere with the Innate Host Defense to Infection NCOMMS-19-18094A

Fig. 3a with significant genes only ($p < 0.05$).

Fig. 4a with significant genes only ($p < 0.05$).

REVIEWERS' COMMENTS:

Reviewer #1 (Remarks to the Author):

Overall I am happy with the changes made. While there are lingering questions, much of this is due to the state of the field - these are things that can be addressed in the future and not significant enough at this stage for me to hold up the timely publication of this important study.

I really enjoyed this work. The authors should be very proud of their accomplishments.

Reviewer #2 (Remarks to the Author):

Congratulations to the authors for this well conducted study. All my concerns have been addressed in this much improved version.

Nelson Gekara